# Molecular Mechanisms Regulating Phenylpropanoid Metabolism in Exogenously-Sprayed Ethylene Forage Ramie Based on Transcriptomic and Metabolomic Analyses

**DOI:** 10.3390/plants12223899

**Published:** 2023-11-18

**Authors:** Hongdong Jie, Pengliang He, Long Zhao, Yushen Ma, Yucheng Jie

**Affiliations:** 1College of Agronomy, Hunan Agricultural University, Changsha 410128, China; jhd20210218@stu.hunau.edu.cn (H.J.); hpl888@stu.hunau.edu.cn (P.H.); azlhh@stu.hunau.edu.cn (L.Z.); mys9204@stu.hunau.edu.cn (Y.M.); 2Hunan Provincial Engineering Technology Research Center for Grass Crop Germplasm Innovation and Utilization, Changsha 410128, China

**Keywords:** ethephon, transcriptome, metabolome, flavonoids, lignin, chlorogenic acid

## Abstract

Ramie (*Boehmeria nivea* [L.] Gaud.), a nutritious animal feed, is rich in protein and produces a variety of secondary metabolites that increase its palatability and functional composition. Ethylene (ETH) is an important plant hormone that regulates the growth and development of various crops. In this study, we investigated the impact of ETH sprays on the growth and metabolism of forage ramie. We explored the mechanism of ETH regulation on the growth and secondary metabolites of forage ramie using transcriptomic and metabolomic analyses. Spraying ramie with ETH elevated the contents of flavonoids and chlorogenic acid and decreased the lignin content in the leaves and stems. A total of 1076 differentially expressed genes (DEGs) and 51 differentially expressed metabolites (DEMs) were identified in the leaves, and 344 DEGs and 55 DEMs were identified in the stems. The DEGs that affect phenylpropanoid metabolism, including *BGLU41*, *LCT*, *PER63*, *PER42*, *PER12*, *PER10*, *POD*, *BAHD1*, *SHT*, and *At4g26220* were significantly upregulated in the leaves. Ethylene sprays downregulated tyrosine and chlorogenic acid (3-*O*-caffeoylquinic acid) in the leaves, but lignin biosynthesis *HCT* genes, including *ACT*, *BAHD1*, and *SHT*, were up- and downregulated. These changes in expression may ultimately reduce lignin biosynthesis. In addition, the upregulation of caffeoyl CoA-*O*-methyltransferase (CCoAOMT) may have increased the abundance of its flavonoids. Ethylene significantly downregulated metabolites, affecting phenylpropanoid metabolism in the stems. The differential 4CL and HCT metabolites were downregulated, namely, phenylalanine and tyrosine. Additionally, ETH upregulated 2-hydroxycinnamic acid and the cinnamyl hydroxyl derivatives (caffeic acid and *p*-coumaric acid). Cinnamic acid is a crucial intermediate in the shikimic acid pathway, which serves as a precursor for the biosynthesis of flavonoids and lignin. The ETH-decreased gene expression and metabolite alteration reduced the lignin levels in the stem. Moreover, the *HCT* downregulation may explain the inhibited lignin biosynthesis to promote flavonoid biosynthesis. In conclusion, external ETH application can effectively reduce lignin contents and increase the secondary metabolites of ramie without affecting its growth and development. These results provide candidate genes for improving ramie and offer theoretical and practical guidance for cultivating ramie for forage.

## 1. Introduction

Ramie, also known as (*Boehmeria nivea* [L.] Gaud.), is a perennial herb in the Urticaceae (nettle) family. Ramie is an excellent phloem fiber crop and an important raw material for feed [1]. The crop can be harvested and utilized several times within a suitable reproductive period. From the second year of new planting, ramie can be harvested eight to ten times a year, and it is generally approximately 70 cm high at harvest. Studies have shown that fresh ramie leaves and stems are rich in nutrients and an excellent source of crude protein, lysine, and calcium in animal feed [2].

Moreover, ramie can improve meat quality, milk fat, and dry matter digestibility [3,4]. Ramie forage serves as an excellent source of plant-based protein feed. Forage ramie is harvested when the plants are young, and the whole plant is typically used as feed. Although the content of lignin in young ramie is comparatively low, it still improves the flavor and digestibility of ramie. Therefore, this study searched for a method to appropriately reduce the lignin content in ramie plants while increasing the secondary metabolites such as flavonoids. These key factors affect the quality of ramie for forage. Flavonoids are polyphenolic compounds primarily found as glycosides in plants. They possess various bioactive properties, such as antioxidant, anti-inflammatory, antibacterial, and anti-tumor effects. These properties play a significant role in maintaining animal health and production [5].

Phenylpropanoid metabolism is a crucial plant metabolic pathway, producing over 8000 metabolites of lignin, flavonoids, and lignans [6]. Lignin is a complex phenylpropanoid polymer and a major component of plant cell walls, which produces high mechanical strength in plant cells. In plants, lignin regulates water transport and resistance to adverse external environments [7]. High lignin contents in forage plants negatively impact the nutrient digestion and absorption in ruminants, consequently reducing the nutritional value of forage and fodder crops. Thus, it is beneficial to regulate the lignin content in plants and increase the levels of secondary metabolites, such as flavonoids, without significantly disrupting plant growth.

Furthermore, understanding the significance of hormones in the overall functioning of plants is crucial for agricultural production [8]. Plant hormones are essential for regulating plant growth and development. Ethylene (ETH) is a precursor that regulates crop growth by controlling multiple aspects of plant growth and development [9]. Ethylene stimulates cell division, promotes growth, induces cell differentiation, and enhances sucrose biosynthesis in the meristematic tissues of sugarcane (*Saccharum officinale* L.) [10]. External ETH application induces the production of phenolic compounds in carrot (*Daucus carota* L.) and lettuce (*Lactuca sativa* L.) [11]. Applying ETH alone or in combination with other stresses can increase the levels of nutrients in plant tissues. Utilizing ETH, independently or in conjunction with other stressors, can enhance nutrient levels in plant tissues. Ethylene can impact the metabolism of numerous plants and consequently influence their growth [12]. The application of ETH has been documented to stimulate the production of additional secondary metabolites in plants [13].

Ethylene-induced transcriptional alterations change the activities of peroxidase (POD) or catalase (CAT) [14]. Phenylalanine is catalyzed by phenylalanine ammonia lyase (PAL) to produce *trans*-cinnamic acid, which is further catalyzed by cinnamate 4-hydroxylase (C4H) to form hydroxycinnamic acid. Hydroxycinnamic acids act as a precursor for the biosynthesis of *p*-coumaric acid, ferulic acid, and caffeic acid, which are vital components for biosynthetic lignin production. The enzyme 4-coumarate: CoA ligase (4CL) catalyzes the biosynthesis of coenzyme A thioesters from these acids, ultimately producing flavonoids [15]. The precursors of phenylpropanoid secondary metabolites and lignin monomers share the same source, suggesting that hormonal regulation of the lignin metabolic pathway can influence the synthesis of secondary metabolites that follow a pathway similar to that of phenylpropanoids. Numerous studies have confirmed that ETH enhances flavonoid biosynthesis by increasing PAL activity [16]. Moreover, ETH enhances fruit ripening while regulating the expression of genes related to flavonoid metabolism [17]. The application of ethylene glycol externally activates genes involved in anthocyanin biosynthesis in grapes (*Vitis vinifera* L.) [18].

Although the effects of plant hormones on the metabolism and biosynthesis of plant phenylpropanoid are well known, the mechanism of the effect of topical ETH applications on the content of phenylpropanoid compounds in forage ramie remains unclear. Therefore, this study compared the transcriptomic and metabolomic profiles of the metabolic pathways in the control and ETH-treatment groups. Additionally, we analyzed the physiological indicators and the biochemical properties of forage ramie. The results of this study will enhance our comprehension of the phytohormone utilization in regulating the genes and metabolites related to lignin biosynthesis in forage ramie. Ultimately, the results will enhance the quality and nutritional values of this valuable forage plant.

## 2. Results

### 2.1. Effect of ETH on the Phenotypic Changes in Forage Ramie

External ETH significantly increased the plant height and stem thickness compared to the control. In particular, the ETH increased the plant height to 73.7 cm, which is 2.7 cm higher than the height of the control (71 cm) (Figure 1A). Similarly, ETH treatment increased the stem thickness by 0.58 mm (9.26 mm vs. 8.68 mm) (Figure 1B). Therefore, the low ETH concentration promoted the growth of ramie.

### 2.2. Effects of ETH on the Content of Secondary Metabolites in the Leaves and Stems of Forage Ramie

Ethylene treatment significantly decreased the lignin content in the leaves (Figure 2A). In particular, the leaves of ETH-treated plants had lignin contents of 12.07%, which was 29.2% lower than that of the control (15.6%) (Figure 2A). However, hormone treatment did not significantly reduce the lignin content in the stem (Figure 2B). The lignin content in the hormone-treated stems was 17.18%, 1.9% lower than that of the control (17.51%). The total flavonoid content of the leaves in the hormone-treated groups was significantly different. The total flavonoid content of the hormone-treated group was 6.62 mg g^−1^, which is 8% higher than the control group (7.15 mg g^−1^) (Figure 2C). Similarly, the total flavonoid content of the group in the stems of the hormone-treated group was 6.28 mg g^−1^, a 26% increase compared to the control (4.98 mg g^−1^) (Figure 2D). Hormone treatment increased the chlorogenic acid content of the leaves (7.89 mg g^−1^) by 5.4% compared to the control (7.48 mg g^−1^) (Figure 2E). Furthermore, ETH increased the hormone chlorogenic acid content of the stems by 16% (4.27 mg g^−1^) compared to the control (3.68 mg g^−1^) (Figure 2F). Thus, ETH use before the harvest of ramie can dramatically increase the accumulation of secondary metabolites, including chlorogenic acid and flavonoids, in the leaves and stems. Additionally, ETH treatment can decrease the lignin content in the stems and leaves of ramie, potentially affecting the nutrient digestion and absorption of ruminants that consume forage ramie.

### 2.3. Metabolomic Data Analysis

#### 2.3.1. The Influence of Ethylene on Variations in the Metabolites of Various Regions of Forage Ramie

Metabolomic analyses were conducted on two specific tissues, namely, the leaves and stems after ETH spray to investigate the impact of ETH on the distinct alterations of metabolites in various sections of forage ramie. Qualitative and quantitative analyses were conducted on the low molecular weight metabolites present in the biological samples. The goal was to identify biologically significant DEMs that varied significantly across the different groups. Principal component analysis (PCA) score plots show the DEMs between the ETH-treated samples and controls in the two tissues of ramie (Figure 3). The variability of PCA for the leaves and stems was 45.9 and 41.2%, respectively.

A total of 51 differential metabolites were obtained in CK-Y vs. A-Y and 55 in CK-J vs. A-J. The metabolites and samples clustered separately based on their abundances (Figure 4), revealing the 13 upregulated metabolites and the 38 downregulated in CK-Y vs. A-Y (Appendix A). CK-J vs. A-J had 17 upregulated and 38 downregulated metabolites (Appendix A).

#### 2.3.2. Analysis of KEGG Enrichment for the Differentially Expressed Metabolites

The study investigated the role of ETH in the growth of forage ramie by conducting an enrichment analysis of the KEGG pathway (Figure 5). The analysis revealed that the DEMs in leaves primarily enriched six pathways, including “Arginine and proline metabolism”, “Glucosinolate biosynthesis”, “Carbapenem biosynthesis”, “Flavone and flavonol biosynthesis”, “Aminoacyl-tRNA biosynthesis”, and “Biosynthesis of unsaturated fatty acids” (Figure 5A, Appendix A). These metabolic pathways, containing the most abundant DEMs, probably play a direct physiological role in enhancing the production of secondary metabolites and influencing lignin metabolism in plants. Similarly, the DEMs in the stems primarily enriched six pathways, including “Valine, leucine, and isoleucine degradation”, “Aminoacyl-tRNA biosynthesis”, “ABC transporters”, “Glucosinolate biosynthesis”, “Cyanoamino acid metabolism”, and “Linoleic acid metabolism” (Figure 5B, Appendix A). The DEMs from both organs of forage ramie significantly enriched phenylpropanoid metabolism. Therefore, external ETH application significantly increased the biosynthesis of flavonoids, flavonols, and other secondary metabolites in ramie, activating key enzymes for lignin biosynthesis and producing anthocyanins, flavonoids, and isoflavones. Thus, ETH reduces the lignin content and increases that of secondary plant metabolites, such as flavonoids.

### 2.4. Transcriptome Analysis of Ramie

#### 2.4.1. Transcriptome Sequencing and Assembly

The samples from group A were treated with an ETH, and the control group CK was sequenced using an Illumina NovaSeq platform (Illumina, San Diego, CA, USA). The leaf group A-Y treated with ETH produced 57.8–84.7 billion bp of raw sequencing data with three replicates. The stem group A-J treated with ETH produced 55.8–72.6 billion bp of raw sequencing data with three replicates. The CK leaf group obtained 56.7–70.5 billion bp of raw sequencing data, while the CK stem group obtained 57.8–74.9 billion bp. After filtering, the ETH-treated leaf and stem groups produced 57.4–84.2 and 55.5–72.1 billion high-quality filtered reads, respectively. The CK leaf and stem groups produced 74.5–84.9 and 57.4–74.44 billion high-quality filtered reads, respectively. In all the groups, the Q20 score exceeded 97%, the Q30 score exceeded 92%, and the average GC content for both the CK- and ETH-treated groups exceeded 47%. The results indicate that the DNA was of high quality and suitable for further analysis (Appendix A).

#### 2.4.2. Analysis of the Differentially Expressed Genes (DEGs) 

A gene is considered differentially expressed if *p* < 0.05 and the absolute value of its log_2_ (fold change) > 2, indicating differential expression between the control and hormone treatment. In the leaf samples of forage ramie, 1076 DEGs were detected (748 upregulated and 328 downregulated). A total of 344 DEGs (107 upregulated genes and 237 downregulated) were identified in the stem samples (Appendix A). The levels of expression of these genes showed significant differences in both the stems and leaves of ramie, indicating their potential significance in the biosynthesis of lignin and flavonoids.

#### 2.4.3. GO Function and KEGG Pathway Gene Enrichment Analysis of the DEGs

The GO (gene ontology) functional enrichment analysis revealed that CK-Y significantly enriched the top five GO terms, including extracellular region (GO:0005576) cell wall organization or biogenesis (GO:0071554), hydrolase activity, acting on glycosyl bonds (GO:0004553), hydrolase activity, hydrolyzing O-glycosyl compounds (GO:0016762), apoplast (GO:0048046) (Figure 6A, Appendix A). The top five GO terms that were significantly enriched in the stems of forage ramie CK-J and A-J are transcription factor activity, sequence-specific DNA binding (GO:0003700), nucleic acid binding transcription factor activity (GO:0001071), RNA polymerase II regulatory region sequence-specific DNA binding (GO:0000977), RNA polymerase II regulatory region DNA binding (GO: 0001012), shoot system development (GO: 0048367) (Figure 6B, Appendix A). Thus, the most important GOs in leaf and stem tissues are completely different before and after classification.

The analysis revealed that both comparison groups significantly enriched the biosynthetic pathways for the secondary metabolites. The significantly enriched pathways differed between the leaf and stem tissues. In the leaf, amino acid sugar and nucleotide sugar metabolism, isoquinoline alkaloid biosynthesis, metabolic pathways, tyrosine metabolism, secondary metabolite biosynthesis, tropane, piperidine and pyridine alkaloids biosynthesis, cyanogenic amino acid metabolism, photosynthesis-antenna protein, starch and sucrose metabolism, and steroid biosynthesis were the most significantly enriched pathways (Figure 7A, Appendix A). Alternatively, the most enriched pathways in stem cells were secondary metabolite biosynthesis, phenylpropanoid biosynthesis, brassinosteroid biosynthesis, ouabain and other terpenoid quinone biosynthesis, fatty acid elongation, isoflavonoid biosynthesis, unsaturated fatty acid biosynthesis, sulfur metabolism, and glycerolipid metabolism (Figure 7B, Appendix A). Both the leaves and stems biosynthesize phenylpropanoids and flavonoids. Thus, we hypothesized that ethylene glycol affects the response of DEGs from these two metabolic pathways to ETH, significantly influencing lignin synthesis in plants. The Appendix A demonstrates the differential expression of genes involved in phenylpropanoid and flavonoid biosynthesis in both the leaves and stems (Appendix A).

#### 2.4.4. Validation of qRT-PCR

We conducted a quantitative real-time reverse transcription PCR (qRT-PCR) of nine crucial DEGs to verify the accuracy of the DEG data acquired from the RNA sequencing (Figure 8). And these results showed that three genes (Bni03G004009, Bni03G005085, and Bni14G018380) were upregulated, whereas six genes (Bni12G016552, Bni07G011041, Bni08G012043, Bni05G008212, Bni10G014837, and Bni13G017586) were downregulated. The outcomes demonstrated a relatively consistent correlation between the qRT-PCR data and the DEG data. The fluorescent-labeled quantitative primers for the DEGs and *Boehmeria nivea* Actin [*BnActin*] (internal control) were designed using the NCBI website (https://www.ncbi.nlm.nih.gov/; accessed on 21 May 2023) (Appendix A). The relative levels of expression were determined as described by Livak et al. [19].

### 2.5. Integrative Metabolomic and Transcriptomic Analysis

The transcriptomic analysis showed that the hormone treatment significantly enriched multiple pathways in the leaf tissues. The pathways encompassed “Amino sugar and nucleotide sugar metabolism”, “Steroid biosynthesis”, “Metabolic pathways”, “Starch and sucrose metabolism”, “Isoquinoline alkaloid biosynthesis”, “Biosynthesis of secondary metabolites”, “Tyrosine metabolism”, “Tropane, piperidine, and pyridine alkaloid biosynthesis”, “Cyanoamino acid metabolism”, and “Photosynthesis—antenna proteins”. In the stems, the hormone treatment enriched several notable pathways, including “Biosynthesis of unsaturated fatty acids, brassinosteroids, phenylpropanoids, ubiquinone, and other terpenoid-quinones”, “Fatty acid elongation and secondary metabolites”, “Isoflavonoid biosynthesis”, and “Sulfur metabolism”.

The metabolomic data revealed 55 and 51 distinct metabolites that enriched various metabolic pathways in the stems and leaves. The phenylpropanoid metabolic pathway was the most abundant in the leaves and stems. Furthermore, a comprehensive analysis of the metabolic and transcriptomic profiles revealed significant differences in the transcriptome and metabolome between the leaves and stems. The DEGs and DEMs that enrich the phenylpropanoid metabolic pathway in the stems and leaves differed. Thus, the different genes and metabolites that enrich the shared phenylpropanoid pathway of flavonoid and phenylpropanoid metabolism are crucial in response to the hormone treatment. Overall, the DEGs of phenylpropanoid metabolism involved several significantly upregulated genes, such as *PER63, PER42, PER12, PER10, POD*, and *At4g26220,* in ramie leaves exogenously sprayed with ETH. This treatment favored the final formation of lignin monomers. 

However, some genes in ramie leaves were significantly upregulated before or at intermediate positions in this process. An example is the upregulation of *BGLU41*, an *Lct* gene that upregulates coumarinate (Coumaninate), which may be important for downregulating lignin. Hormone treatment also upregulated other genes such as *BAHD1* and *SHT*. The downregulation of *ACT* produced the four DEMs: caffeoylshikimic acid, caffeoylquinic acid, coumaroyl CoA, and caffeoyl CoA. The caffeoyl CoA-*O*-methyltransferase (COMT) products may increase the abundance of its flavonoid metabolites (Figure 9). Although the DEGs that affected phenylalanine metabolism, including 4*CL,* were significantly downregulated in the stems, the DEMs of HCT, including phenylalanine and tyrosine, were downregulated. 2-Hydroxycinnamic acid was upregulated in the stems, and cinnamyl hydroxyl derivatives included *p*-coumaric acid and caffeic acid. 

Cinnamic acid is a crucial and pivotal component of the mangiferous acid pathway and is a precursor for flavonoid or lignin biosynthesis. The downregulation of these specific genes and their corresponding metabolites decreased the lignin levels within the stem, and *HCT* downregulation probably inhibited lignin biosynthesis to promote flavonoid biosynthesis (Figure 10).

## 3. Discussion

The rapid development of animal husbandry is increasing the shortage of raw materials for feed preparation [20]. Ramie is an important fiber cash crop in southern China for its role in producing fibers [21,22,23], and the forage value of ramie has been the subject of recent studies. However, limited research has been conducted on the enzymatic activities linked to the biosynthesis of ramie lignin using ethylene glycol. Forage ramie is harvested when the plant is young, and the whole plant is typically fed to the animals. Although the content of lignin in young China grass is comparatively low, it can still impact the flavor and digestibility of this feed for animals. High lignin content adversely affects the digestion of ramie forage by animals and the utilization of its bast fibers. The presence of lignin in the feeds supplemented with forage impacts the digestion and absorption of forage crops by ruminants, decreasing the nutritional value of both forage and forage crops [24]. The reduction in dry matter digestibility of forage directly reduces its nutritional quality [25]. Therefore, this study examined how ETH application changes secondary metabolism in the leaves and stems of ramie to improve the products with valuable feed ingredients. Ethylene enhances secondary metabolite accumulation, accelerates enzyme catalysis, and promotes the formation of bioactive compounds, such as alkaloids, flavonoids, terpenoids, and polyphenolics [26]. We hypothesized that ETH enhances the accumulation of secondary metabolites in ramie. The findings demonstrated that low ETH concentrations stimulate the growth of ramie, supporting our hypothesis. We conducted tests to determine the levels of lignin, total chlorogenic acid, and total flavonoids in the leaves and stems of forage ramie. The results showed a significant reduction in lignin content in the leaves of the group treated with the hormone (Figure 2A).

The lignin content in the leaf blades of the hormone-treated group was 12.07%, 29.2% lower than the content of the control group (15.6%) (Figure 2A). In contrast, ETH treatment did not significantly reduce the lignin in the stems (Figure 2B). The stem of the hormone-treated group had 17.18% lignin, 1.9% lower than the control (17.51%). The total flavonoid content of the leaves in the hormone-treated group increased but did not reach a statistically significant level. The leaf group treated with hormone had a flavonoid content of 6.62 mg g^−1^, which was 8% higher than the control group (7.15 mg g^−1^) (Figure 2C).

Similarly, the stem group treated with hormone had 6.28 mg g^−1^ of total flavonoids, 26% higher than the control group (4.98 mg g^−1^) (Figure 2D). Hormone treatment increased the chlorogenic acid contents in the leaves treated with hormone by 5.4% (7.89 mg g^−1^) compared to the control group value of 7.48 mg g^−1^ (Figure 2E). Similarly, chlorogenic acid content in the stems treated with hormone increased by 16% (4.27 mg g^−1^). In contrast, the control group only had 3.68 mg g^−1^ of chlorogenic acid (Figure 2F). These data highlight the significance of ETH treatment in promoting the accumulation of secondary metabolites in ramie. The regulation by exogenous ETH increased the secondary metabolites in ramie without affecting its growth and decreased lignin formation. In contrast, ETH increased the contents of flavonoids and the ester chlorogenic acid, owing to the inhibited biosynthesis of lignin after exogenous ETH application [27]. Therefore, ETH treatment accumulated additional secondary metabolites and modified the metabolic genes associated with lignin and flavonoid biosynthesis [27].

In the phenylpropanoid pathway, *p*-coumaryl CoA is at the crossroads of the metabolic pathways that lead to the upregulation of flavonoids or monolipids and sinapoylmalate. The metabolic flux into these two pathways is regulated by the activity of CHS and HCT [28]. CHS is regulated to promote the biosynthesis of flavonoids and is the source of all flavonoids. It is considered a key enzyme in flavonoid biosynthesis [29], and hydroxycinnamoyl CoA mangiferyl/quinic acid hydroxycinnamoyltransferase (HCT) is considered an essential enzyme for regulating lignin biosynthesis and composition [30]. RNA-seq and LC-MS/MS assays were conducted to understand the underlying molecular and metabolic mechanisms by which ETH spraying alters some physiological markers, lowers lignin, elevates secondary metabolites, and promotes secondary metabolite synthesis. The transcriptomic and metabolomic findings of the forage ramie treated with ETH revealed that specific DEGs and DEMs influenced the biosynthesis of metabolites in its leaves and stems. The results of this study indicate a significant upregulation of the DEGs involved in phenylpropanoid metabolism in the leaves after the exogenous ETH application on forage ramie. 

The genes that were upregulated included *BGLU41*, *LCT*, *PER63*, *PER42*, *PER12*, *PER10*, *POD*, *BAHD1*, *SHT*, and *At4g26220*, and the upregulation of these genes resulted in the differential production of the metabolite tyrosine in the leaves. Chlorogenic acid (3-*O*-caffeoylquinic acid) was downregulated, while the up and downregulation of three key genes for lignin synthesis, *HCT* (*ACT*, *BAHD1*, and *SHT*), reduced the biosynthesis of lignin, resulting in low lignin levels. The upregulation of caffeoyl CoA-*O*-methyltransferase may have increased the abundance of its flavonoid metabolites. The upregulation of *At4g26220*, which was first identified in Arabidopsis and is reported to have a strong preference for the methylation of flavanones and dihydroflavonols, probably contributed to the elevated contents of flavonoids in ramie [31]. Some studies have reported the involvement of lignin peroxidase, manganese peroxidase, multifunctional peroxidase, or laccase in the degradation of lignin [32]. 

Laccase and peroxidase can also degrade lignin through the production of low molecular weight radicals, such as those of hydroxyl groups (hydroxyl radical [OH^−^] and superoxide anion [O^2−^]), that depolymerize phenolic and nonphenolic lignin polymers [32], consistent with the results of this study. These molecules could be responsible for the reduced lignin content of ramie. β-glucosidase was upregulated in the leaves, which could result from the gene downregulation in the lignin biosynthetic pathway that led to improvements in the release of sugar [33]. In contrast, the significant downregulation of DEGs that affected phenylalanine metabolism was detected in the stems, including those of *4CL, HCT*, and *CCR*. Its differential metabolites, phenylalanine, and tyrosine were downregulated, 2-hydroxycinnamic acid was upregulated, and the cinnamyl hydroxyl derivatives were *p*-coumaric acid and caffeic acid. Cinnamic acid is a critical intermediate in the mangiferous acid pathway and serves as a precursor for the biosynthesis of flavonoids or lignin. The downregulation of these specific genes and their corresponding metabolites decreased the levels of lignin in the stems. Moreover, the activation of most phenylpropanoid pathway genes by ethylene glycol substantially increased the accumulation of diverse phenylpropanoid-like compounds. Lignin biosynthesis involves several key genes: *4CL, HCT*, and *CCR* [34]. Studies have demonstrated that inhibiting the expression of *4CL* in transgenic poplar (*Populus* sp.) significantly reduces the lignin content [34]. Perennial ryegrass (*Lolium perenne* L.) with a low content of lignin could be obtained by silencing the *CCR* gene [35] and downregulating the *HCT* gene in alfalfa (*Medicago sativa* L.) lowered the lignin content, making it more digestible. This study investigated the transcriptome and metabolome of forage ramie under hormone treatment using association analysis. The objective was to identify the differentially expressed genes and metabolites. The findings from this study will significantly contribute to exploring functional genes. This study systematically explains the mechanisms that regulate lignin in forage ramie and establishes a theoretical foundation for applying ETH.

## 4. Materials and Methods

### 4.1. Collection and Measurement of the Physiological Indicators

The ramie variety selected for this study was Zhong Ramie No. 1, cultivated at the Experimental Base of Hunan Agricultural University, Cultivation Park, China. A total of 15 healthy seedlings were randomly selected and labeled as one treatment. Each treatment was repeated three times, and fertilizer was applied on 27 September 2021. The plants were mowed on this date, and no fertilizer was utilized throughout the growth period. The plant height was measured with a ruler to determine the distance from the stem base to its top in cm. Vernier calipers were used to measure the diameter from the base of the plant to the tip at half of its height to measure the stem thickness in mm.

Ethylene (ETH) white powder was purchased from Beijing Solebro Technology Co. (Beijing, China). The Ethephon agents were diluted to 5 mg L^−1^ before use. Ramie was treated once with ETH 5 days before harvest. The forage ramie was mowed when 80% of the plants had reached 80 cm. We divided the experimental samples into two groups to guarantee the coherence of subsequent correlation analysis. One sample was used to determine the physiological and biochemical indicators, and the other was used for the transcriptome and metabolome analyses.

### 4.2. Quantification of Lignin, Chlorogenic Acid, and Flavonoids

#### 4.2.1. Quantification of Lignin and Total Flavonoids

The dried samples were crushed with a pulverizer (DF-25 pulverizer; Shanghai, China). They were then filtered through a 0.2 mm sieve, stored in sealed bags, and assigned unique numbers. The lignin and total flavonoid contents of these samples were determined using BC4200 and BC1335 assay kits (Solebro, Beijing, China) following the manufacturer’s instructions. The data were presented as the mean ± SD of three independent experiments and analyzed by ANOVA using SPSS 19.0 (IBM, Inc., Armonk, NY, USA).

#### 4.2.2. Determination of the Total Chlorogenic Acid in Ramie

A total of 0.1 g of the crushed sample was placed in a 20 mL calibrated test tube and submerged in 50% ethanol with a pH of 4 (using a solid–liquid ratio of 1:50) for 24 h. After this, the solution was extracted ultrasonically for 30 min at 40 °C. Finally, the mixture was filtered and rinsed thoroughly. The mixture was extracted twice, and the filtrate was combined with 50% ethanol to 10 mL and used as the test sample. A volume of 1 mL of the solution was diluted to an appropriate amount so that the measurements were within the levels of the standard curve. This mixture generally required a 10-fold dilution. The absorbance at 310 nm was measured using 50% ethanol as the blank. Absorbance A was measured and subsequently converted into concentration C (mg mL^−1^) using an equation derived from the standard curve. 

### 4.3. Transcriptome Analysis

A TRIzol kit (Invitrogen, Carlsbad, CA, USA) was used to extract the total RNA. The quality of RNA was assessed using an Agilent 2100 Bioanalyzer (Agilent Technologies, Santa Clarita, CA, USA). 

RNA quality was examined using RNAase-free agarose gel electrophoresis. The rRNA was removed from the total RNA using the Ribo ZeroTM magnetic kit (Epicentre, Madison, WI, USA). The remaining mRNA was enriched and fragmented into shorter fragments using the Fragmentation Buffer. An NEB Next Ultra RNA Library Preparation Kit for Illumina (NEB# 7530, New England Biolabs, Ipswich, MA, USA) was used to prepare the fragmented mRNA for reverse transcription into cDNA. Purified double-stranded cDNA fragments underwent end repair, base addition, and ligation with Illumina sequencing adapters. The ligation reaction was then purified using 1× AMPure XP beads (Beckman Coulter, Inc., Brea, CA, USA). Ligated fragments were size selected through agarose gel electrophoresis, followed by PCR amplification. The resulting cDNA libraries were sequenced using an Illumina NovaSeq 6000 platform. Kidio Biotechnology Co., Ltd. (Guangzhou, China) extracted the RNA, prepared the library, and analyzed the data for high-throughput sequencing. The analysis of RNA sequence data began with examining raw sequencing data using FastQC. Low-quality reads were filtered and discarded using the Trimmomatic (v. 0.36) tool. Contaminated reads with adapter sequences were trimmed, and the resulting clean reads were aligned to the reference genome of the ramie using the STATR software (v. 2.5.3a). The gene expression levels were estimated using the fragment per kilobase of transcript per million mapped reads (FPKM) approach. All the assembled individual genes were annotated in the GO and KEGG databases. The DEGs between the two groups were identified using edgeR (v. 3.12.1).

### 4.4. Metabolite Analysis

#### 4.4.1. Metabolite Extraction and Detection

The leaves and stems of the ETH (A-Y/A-J) and control (CK-Y/CK-J) groups were freeze-dried under vacuum. The samples were then pulverized into a fine powder using a grinder (MM 400, Retsch, Haan, Germany) at 30 Hz for 1.5 min. A total of 100 mL of the product powder was dissolved in 1 mL of extraction solution. The dissolved sample was incubated overnight at 4 °C, and the mixture was vortexed three times. After centrifugation at 10,000× *g* for 10 min, the supernatant was removed, and the sample was filtered through a 0.22 µm microporous membrane. The filtered samples were then transferred to sample vials for LC-MS/MS analysis. The data acquisition instrument system primarily comprised ultra-high performance liquid chromatography (UPLC) (Shimpack UFLC SHIMADZU CBM30A, Shimadzu, Tokyo, Japan) and tandem mass spectrometry (MS/MS) (Applied Biosystems 6500 QTRAP, http://www.appliedbiosystems.com.cn/; accessed on 22 September 2022). The metabolomics sampling utilized the same protocol as the transcriptomic analysis with three biological replicates per treatment.

#### 4.4.2. Pre-Processing of the Data and Identification of the Metabolites

Data filtering, peak detection, alignment, and computation were performed using Analyst 1.6.1 software (SCIEX, Framingham, MA, USA). A manual examination was conducted to identify the peaks with a signal-to-noise ratio (s/n) > 10 to generate a matrix with less bias and redundant information. We used in-house software written in Perl to remove redundant signals caused by different isotopes, internal fragments, K^+^, Na^+^, NH_4_^+^ complexes, and dimers with precise *m*/*z* values were obtained for each Q1 to enhance identifying and annotating metabolites. Total ion chromatograms (TICs) and extracted ion chromatograms (EICs or XICs) were generated from the quality control (QC) samples to obtain a comprehensive understanding of the metabolite profiles in all the samples. The calculations were conducted to ascertain the area of every chromatographic peak. The peaks were subsequently categorized into different samples based on spectral patterns and retention times. We searched internal and public databases, MassBank, KNApSAcK, HMDB, MoTo DB, and METLIN, for comprehensive metabolite identification [36]. The *m*/*z* values, retention time (RT), and fragmentation patterns were compared with the standards for identification. According to the variable importance in projection (VIP) scores in the (O)PLS model, the metabolites that exhibited the most significant differences between the two groups were prioritized for sorting. The VIP threshold for log implication was set at 1. Additionally, a *t*-test was used as a univariate analysis to identify unique metabolites. Metabolites with a *t*-test *p*-value < 0.05 and VIP ≥ 1 were considered DEMs between the two groups. KEGG is the main publicly accessible database for pathways. The KEGG database analyzes genes and metabolites [37]. Thus, the metabolites were compared with the KEGG metabolic pathways, and subsequent enrichment analyses were performed accordingly. Pathway enrichment analysis identifies significantly enriched metabolic or signal transduction pathways in DEMs.

### 4.5. Validation of Quantitative Real-Time Reverse Transcription Polymerase Chain Reaction (qRT-PCR)

Nine differentially expressed genes were selected to verify the reliability of the transcriptome data. Each reaction system contained 20 µL, comprising 10 µL of SYBR greenMasterMix, 0.4 µL of forward and reverse primers, 2 µL of cDNA, and 7.2 µL of ddH2O. The specific primers were designed using Primer5.0 software. The amplification conditions were as follows: initial denaturation at 95 °C for 15 s, 60 °C for 30 s, 72 °C for 30 s, and 40 cycles. The relative expression of each gene was calculated using the 2^-DDCT^ method, with actin as the internal reference gene. The primer sequences are shown in Appendix A.

### 4.6. Statistical Analysis

The data were presented in the form of mean ± standard error. An ANOVA was conducted using SPSS 26.0 (IBM, Inc., Armonk, NY, USA) to analyze the data, followed by post hoc tests to detect significant differences.

## 5. Conclusions

This study compared field traits, physiological and biochemical indicators, and the transcriptome and metabolome of forage China grass treated with ETH-treated control to reveal the mechanism underlying plant response to hormone regulation. Ethylene reduces the content of lignin to increase the number of secondary metabolites, such as flavonoids. Inhibiting lignin biosynthesis redirects the metabolic fluxes to flavonoid biosynthesis. This study contributes to the functional analysis of the key genes involved in lignin biosynthesis in ramie and provides a theoretical foundation for breeding forage China grass and promoting the comprehensive utilization of its varieties.

## Figures and Tables

**Figure 1 plants-12-03899-f001:**
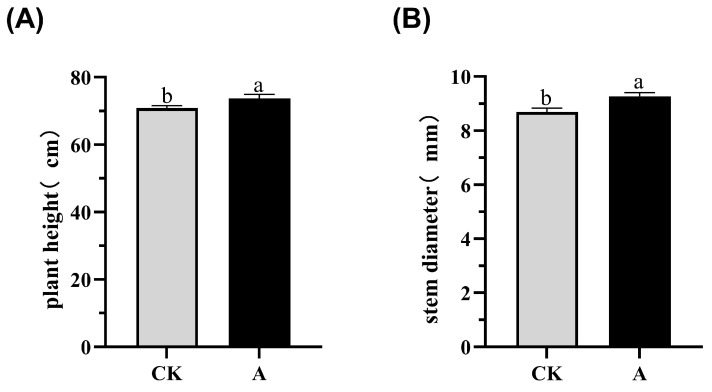
The effects of ETH treatment on the height and stem thickness of forage ramie. CK, the control group (no treatment). A, the ETH treatment group. The plant height is labeled as (**A**) and the stem thickness as (**B**). The mean ± SD (n = 3) is represented on the vertical axis. Significant differences at *p* < 0.05 are indicated by different letters.

**Figure 2 plants-12-03899-f002:**
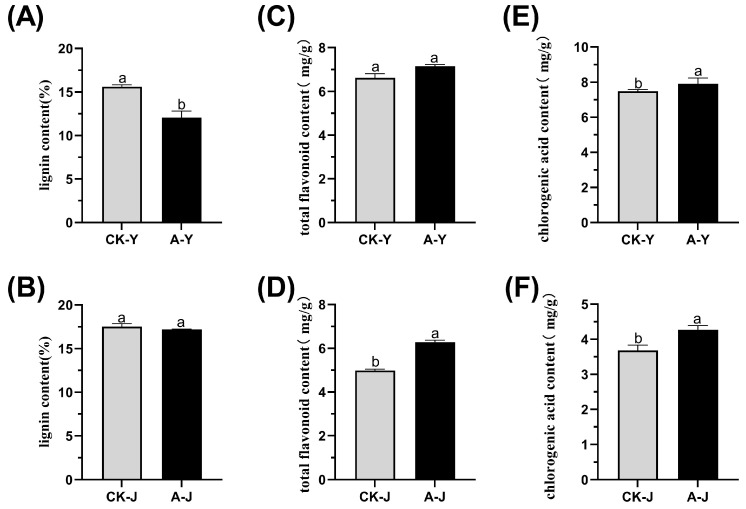
The effect of ETH treatment on the levels of lignin, total flavonoids, and chlorogenic acid in the leaves and stems of forage ramie. CK-Y, the control leaves; CK-J, the control stems; A-Y, the leaves treated with ETH; A-J, the stems treated with ETH. (**A**) Lignin content in the leaves. (**B**) Lignin content in the stems. (**C**) Total flavonoid content in the leaves. (**D**) Total flavonoid content in the stems. (**E**,**F**) Chlorogenic acid content in the leaves and stems. The mean ± SD (n = 3) is represented on the vertical axis. Significant differences at *p* < 0.05 are indicated by different letters.

**Figure 3 plants-12-03899-f003:**
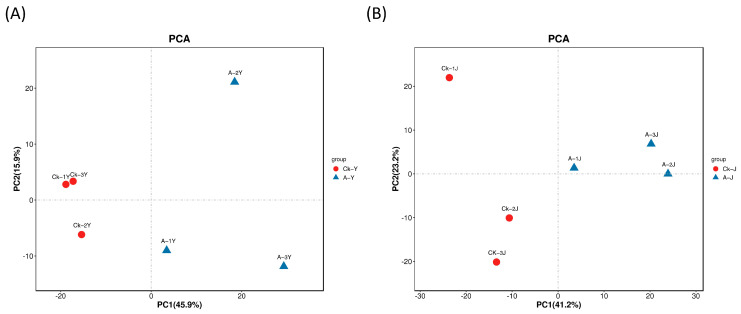
Principal component analysis (PCA) for ramie metabolites. (**A**) PCA analysis of two comparison groups in the leaves (Y). (**B**) PCA analysis of two comparison groups in the stems (J). Red, CK-Y, CK-J. Blue, A-Y, A-J. A-J, leaves treated with ETH; A-Y, stems treated with ETH; CK-J, control leaves; CK-Y, control stems.

**Figure 4 plants-12-03899-f004:**
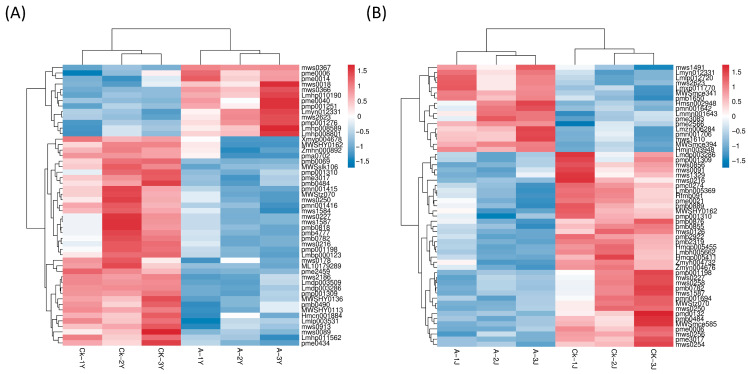
Heat map illustrating the thermal clustering of differentially expressed metabolites. Red, metabolites that were highly abundant. Blue, metabolites that have low levels. (**A**) CK-Y vs. A-Y comparison group. (**B**) CK-J vs. A-J comparison group. A-J, stems treated with ETH; A-Y, leaves treated with ETH; CK-J, control stems; CK-Y, control leaves.

**Figure 5 plants-12-03899-f005:**
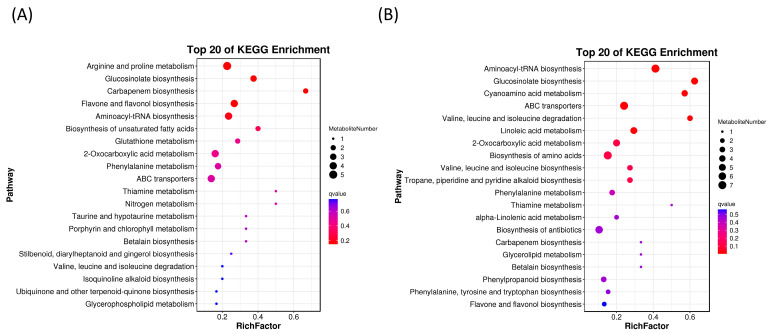
(**A**) A KEGG analysis on the leaf group. (**B**) KEGG analysis on the stem group. The vertical axis represents pathways, and the horizontal axis represents enrichment factors (the ratio of differential genes to the total number of genes in the pathway). The size of markers indicates the number of genes, with more intense red indicating a smaller Q value. A-J, stems treated with ETH; A-Y, leaves treated with ETH; CK-J, control stems; CK-Y, control leaves; KEGG, Kyoto Encyclopedia of Genes and Genomes.

**Figure 6 plants-12-03899-f006:**
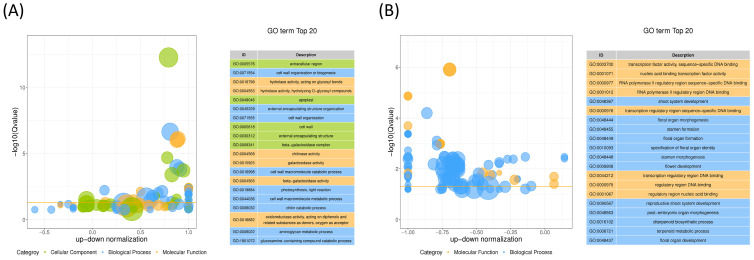
GO functional enrichment analysis for differentially expressed genes in the stems and leaves of ramie after ETH treatment. (**A**) Top 20 GO terms in CK-Y vs. A-Y. (**B**) Top 20 GO terms in CK-J vs. A-J. Difference bubbles in the GO enrichment plot illustrate the relationship between the log_10_ (Q value) on the vertical axis and the z-score value on the horizontal axis. Z-score, the difference between the number of upregulated and downregulated differential genes as a percentage of the total DEGs. Yellow line, the threshold of Q value = 0.05. The list of GO terms on the right is the top 20 Q values. Different colors represent different ontologies, and the subclasses are plotted separately for each ontology. A-J, stems treated with ETH; A-Y, leaves treated with ETH; CK-J, control stems; CK-Y, control leaves; GO, gene ontology.

**Figure 7 plants-12-03899-f007:**
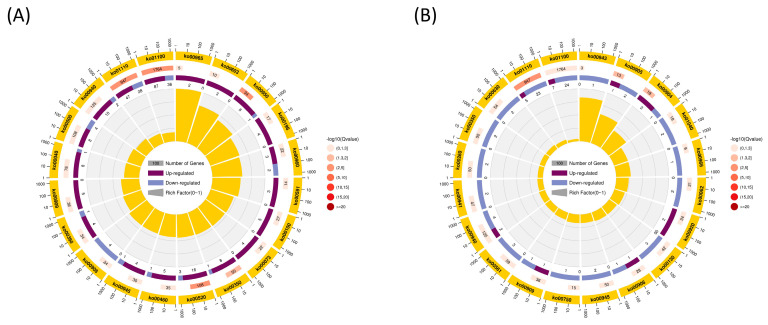
KEGG pathway enrichment of differentially expressed genes (DEGs) in the stems and leaves of ETH-treated ramie. (**A**) Enriched KEGG pathways of DEGs between CK-Y and B-Y. (**B**) Enriched KEGG pathways of DEGs between CK-J and B-J. Different colors in the circles represent the top 20 enriched pathways, the number of DEGs, the fold change in DEGs, and the rich factor value for each pathway.

**Figure 8 plants-12-03899-f008:**
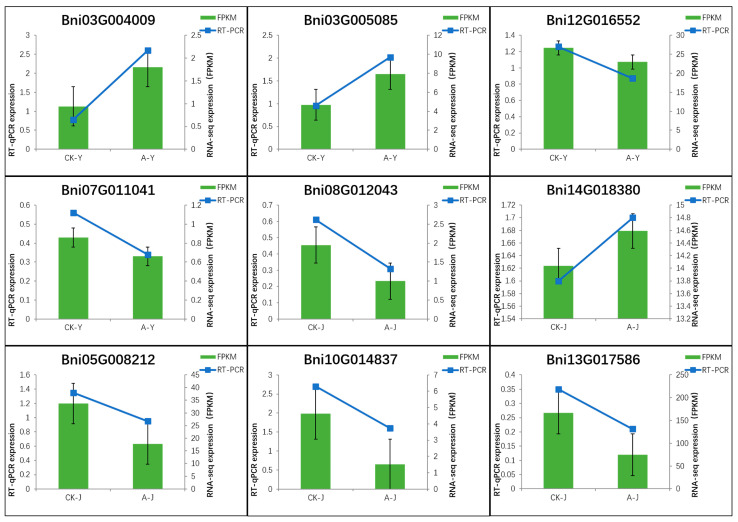
qRT-PCR of nine key genes in the phenylpropanoid biosynthetic pathway. FPKM, fragments per kilobase of transcript per million mapped reads; qRT-PCR, quantitative real-time reverse transcription PCR.

**Figure 9 plants-12-03899-f009:**
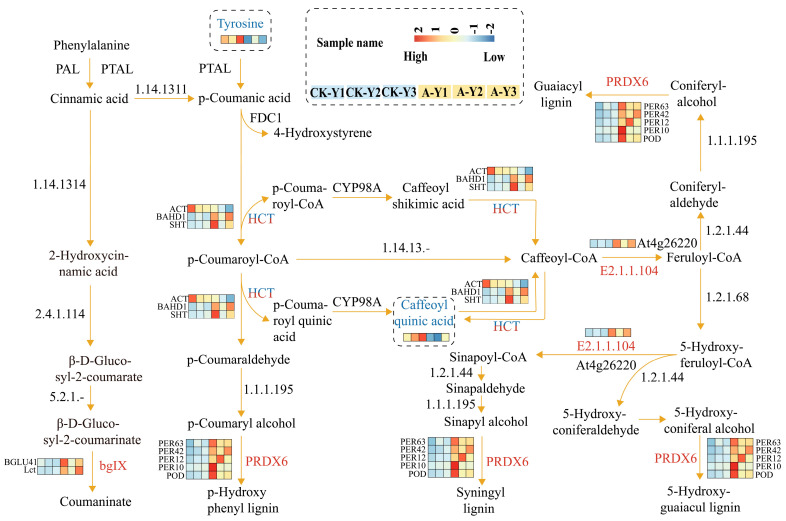
A schematic diagram showing the phenylpropanoid biosynthesis pathway in ramie leaves. In this scheme, different metabolites are displayed within dashed boxes that form a heat map. Differentially expressed genes are shown next to their corresponding enzymes as heat maps. The gene scale corresponds to the relative levels of transcription of the enzymes.

**Figure 10 plants-12-03899-f010:**
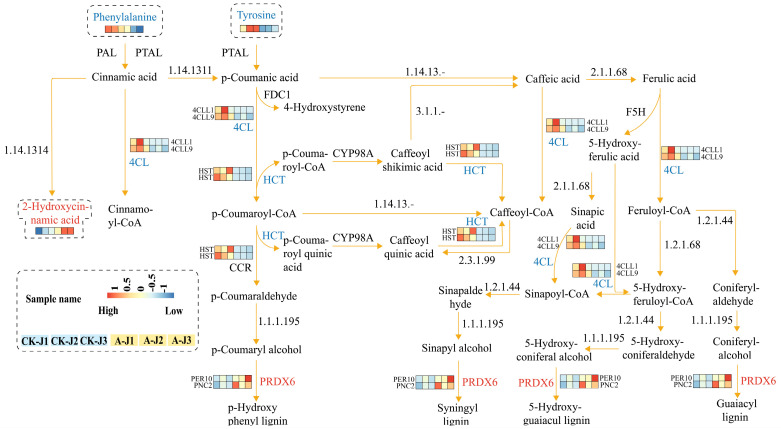
Schematic representation of the phenylpropanoid biosynthetic pathway in ramie stems. In this case, the different metabolites are shown as a heat map in dashed boxes. Differentially expressed genes are indicated alongside their corresponding enzymes (presented as heat maps). The scale of the genes corresponds to the relative transcript level range of the enzymes.

## Data Availability

Data are contained within the article and Appendix A.

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
