# Peer review of "Molecular Mechanisms Regulating Phenylpropanoid Metabolism in Exogenously-Sprayed Ethylene Forage Ramie Based on Transcriptomic and Metabolomic Analyses"

_plants, 2023, doi:10.3390/plants12223899_

Round 1

Reviewer 1 Report

Comments and Suggestions for Authors

The authors of the manuscript consider the possibilities of improving some of the characteristics that determine the quality of food in the ramie plant using modern methods of transcriptomic and metabolomic analysis. Ramie is considered as a promising plant used to produce high-quality animal feed. However, this type of plant has a significant drawback that requires correction by researchers, namely, the increased lignin content. It is known that high lignin content in plants used for animal feed leads to undesirable consequences. In this regard, finding ways to reduce it in plants used for feed purposes seems to be an extremely urgent task. For this, the authors propose a rather original approach - modification of secondary metabolism with redistribution of metabolites towards a decrease in lignin and an increase in flavonoids in ramie plants under the influence of ethylene applied to the plants in the form of a spray. The use of this technique is not in doubt, since the slowdown of the metabolism of ripening fruits as a result of exogenous exposure to ethylene is known, and the technique of treating plants in the form of dsDNA sprays to induce certain pathways of biosynthesis of secondary metabolites is also known. In this manuscript, the authors conduct a comparative analysis of the results of transcriptomic and metabolomic analyzes with the presentation of the results in two diagrams presented in Figures 9 and 10, which opens up great opportunities for the development of further work using modern methods of molecular analysis. For example, identifying key genes for lignin biosynthesis will serve as the basis for work on genome editing of ramie plants to improve its nutritional characteristics. The authors used generally accepted methods for analyzing the obtained data using publicly available bioinformatics databases and programs. The results are correctly presented in the figures. This manuscript may be recommended for publication in its presented form. 

Reviewer 2 Report

Comments and Suggestions for Authors

This current paper by Hongdong Jie.  At al, crop Ramie (Boehmeria nivea [L.] Gaud.), was studied regarding the impact of ETH sprays on the growth and metabolism of forage ramie, and explored the mechanism of ETH regulation on the growth and secondary metabolites of forage ramie in combination with transcriptomic and metabolomic analyses. A total of 1,076 differentially expressed genes (DEGs) and 51 differentially expressed metabolites (DEMs) were identified in the leaves, and 344 DEGs and 55 DEMs were identified in the stems. T.

 comments

paper is sound since try to cover the underlying molecular and metabolic mechanisms by which ethylene spraying  alters some physiological markers, lowers lignin, elevates secondary metabolites, and promotes secondary metabolites productions in this important crop

organization of the paper, good results test diagrams and figures makes the paper very good

please check English language

it would be good to quantify main phenolics and validate the procedure according to ICH guidelines including recovery, etc

Comments on the Quality of English Language

some reparation of english needed

Reviewer 3 Report

Comments and Suggestions for Authors

Jie and his (her) colleagues aim to explore the mechanism of ETH regulation on the growth and secondary metabolites of forage ramie in combination with transcriptomic and metabolomics analyses. The manuscript enriches our knowledge about the phenylpropanoid metabolism in forage ramie. The submitted manuscript is written clearly and general interest to the readers. However, I have several comments that should be addressed before publication.

In scientific aspects, I have some comments:

1.       The results of the qRT-PCR of nine crucial DEGs need further analysis in the result part. The authors only present a figure8.

2.       Although the author has drawn schematic diagrams displayed the biosynthetic pathway of phenylpropanoids in the leaves and stems of ramie, the DEGS in the key nodes should be further detected using qRT-PCR in the revised manuscript.

Comments on the Quality of English Language

In language aspect:

1.  The “,” should be inserted before the “is” (Page1, L43), for example, “, is a perennial herb”.

2.  The ‘……plants or crops’ (Page2, L68) is suggested for deleting and remaining one, because crops are kind of plants.

3.  “……the development of color” is not an idiomatic English expression, please check and correct it.

4.  The space should be inserted between the real and time (Page10, L245).

5.  The authors must carefully check grammar, punctuation, spelling, and overall style in the whole text.

Reviewer 4 Report

Comments and Suggestions for Authors

The manuscript submitted by Jie et al. represents an interesting research work.  The experiments were designed properly and the analysis were carried out thoughtfully.  The work showed a significant progress in this area with this important forage plant.

 Some minor revision or clarifications are needed.

1)      Did you measure the biomass of treatments?  Biomass, such as dry weight,  would be a more proper measurement for growth, than height or diameter.  The number of plants used for measuring plant heights and diameters are not clear. 

2)      Line 376, the ethylene agent is not clearly described, is that called Ethephon or ethrel?  Ethylene itself is a gas, cannot be diluted as a liquid, need to be clarified. 

3)      Line 376, “China grass” what is it?

4)      Line 181, “with a hormone” – it is better to use “with ETH or ethylene” through the manuscript.

5)      Line 192-193, for DEG identification, need to provide a reference. Many papers use  log2(fold change) > 2.

6)      Line 200 -201, “Top” should be “top”?

7)      Line 401, “prokaryotic RNA”?  where prokaryotic RNA comes from?

Comments on the Quality of English Language

Round 2

Reviewer 3 Report

Comments and Suggestions for Authors

Jie and his (her) colleagues aim to explore the mechanism of ETH regulation on the growth and secondary metabolites of forage ramie in combination with transcriptomic and metabolomics analyses. The revised manuscript improved obviously. However, some comments suggested by me before have not been considered in the revised manuscript. So I still insist that the results of the qRT-PCR of nine crucial DEGs need further analysis in the result part. The authors only present a figure 8.

Comments on the Quality of English Language

The revised manuscript reads well.
